# Insights into Genetic and Antigenic Characteristics of Influenza A(H1N1)pdm09 Viruses Circulating in Sicily During the Surveillance Season 2023–2024: The Potential Effect on the Seasonal Vaccine Effectiveness

**DOI:** 10.3390/v16101644

**Published:** 2024-10-21

**Authors:** Fabio Tramuto, Carmelo Massimo Maida, Giulia Randazzo, Adriana Previti, Giuseppe Sferlazza, Giorgio Graziano, Claudio Costantino, Walter Mazzucco, Francesco Vitale

**Affiliations:** 1Department of Health Promotion Sciences Maternal and Infant Care, Internal Medicine and Medical Specialties “G. D’Alessandro”-Hygiene Section, University of Palermo, 90127 Palermo, Italy; carmelo.maida@unipa.it (C.M.M.); claudio.costantino01@unipa.it (C.C.); walter.mazzucco@unipa.it (W.M.); francesco.vitale@unipa.it (F.V.); 2Regional Reference Laboratory for Molecular Surveillance of Influenza, Clinical Epidemiology Unit, University Hospital “Paolo Giaccone”, 90127 Palermo, Italy; giulia.randazzo1986@gmail.com (G.R.); adriana.previti@policlinico.pa.it (A.P.); giuseppe.sferlazza@policlinico.pa.it (G.S.); giorgio.graziano@policlinico.pa.it (G.G.)

**Keywords:** influenza virus, molecular surveillance, whole-genome sequencing, antigenic drift, vaccine efficacy, community, Sicily, Italy

## Abstract

After disruption in the influenza circulation due to the emergence of SARS-CoV-2, the intensity of seasonal outbreaks has returned to the pre-pandemic levels. This study aimed to evaluate the evolution and variability of whole-genome sequences of A(H1N1)pdm09, the predominant influenza virus in Sicily (Italy) during the season 2023–2024. The potential vaccine efficacy was calculated using the *p*_epitope_ model based on amino acid changes in the dominant epitope of hemagglutinin. The HA gene sequences showed several amino acid substitutions, some of which were within the major antigenic sites. The phylogenetic analysis showed that Sicilian strains grouped into two main genetic clades (6B.1A.5a.2a.1 and 6B.1A.5a.2a) and several subclades. Notably, about 40% of sequences partially drifted from the WHO-recommended vaccine strain A/Victoria/4897/2022 for the Northern Hemisphere. These sequences mostly belonged to the subclades C.1.8 and C.1.9 and harboured the amino acid mutations responsible for the modest predicted vaccine efficacy (E = 38.12% of 53%, *p*_epitope_ = 0) against these viruses. Amino acid substitutions in other gene segments were also found. Since influenza viruses are constantly evolving, genomic surveillance is crucial in monitoring their molecular evolution and the occurrence of genetic and antigenic changes, and, thus, their potential impact on vaccine efficacy.

## 1. Introduction

Influenza A and B viruses are among the leading causes of respiratory disease worldwide, affecting up to 20% of the general population seasonally [1]. Influenza disease may vary in severity, with pre-existing individual risk factors and comorbidities able to exacerbate its impact, potentially leading to severe complications and deaths [2,3,4,5].

It has been globally reported that the COVID-19 pandemic has significantly disrupted the occurrence and trends of various respiratory pathogens, including influenza viruses [6,7,8]. Consequently, the “immunity debt” [9] that emerged in a large segment of the population has promoted a renowned spread of influenza in the community, returning to pre-pandemic levels worldwide in the last two seasons.

Although with geography-related variations, A(H1N1)pdm09, A(H3N2), and B viruses have been detected again in most countries [10,11], exhibiting seasonally different dynamics in their epidemiology.

Additionally, due to constant viral evolution, circulating influenza strains progressively vary. New variants may emerge or, conversely, may occasionally disappear. In this respect, for example, the B/Yamagata lineage has not been detected to be actively circulating in global surveillance since March 2020 [12].

It is well-known that the influenza virus genome consists of eight different negative-sense, single-stranded RNA segments, named PB2, PB1, PA, NP, HA, NA, MP, and NS, which encode several proteins [13]. Among these, the two surface glycoproteins hemagglutinin (HA) and neuraminidase (NA) play a pivotal role in viral infectivity and pathogenesis, being strictly required to mediate both virus entry and cell-to-cell spread within the host’s airway epithelium [14,15].

Hemagglutinin elicits the primary neutralizing response harbouring the antigenic and receptor-binding sites on the globular head domain (HA1). Among influenza A, five major antigen sites are recognized in A(H1N1)pdm09 (Sa, Sb, Ca1, Ca2, and Cb) and A(H3N2) (A through E) [16], as well as the sialic acid receptor-binding sites (RBSs) [17,18]. Evolution due to natural selection or positive pressure may favour the accumulation of mutation in the HA protein, which may lead to the emergence of influenza virus strains able to escape the existing neutralizing antibodies, thus compromising the efficacy of vaccines.

For this reason, the influenza vaccine composition is reviewed annually based on the WHO’s recommendation [19] or, even more importantly, adapted to the most recent evidence in molecular epidemiology [12,20]. Nevertheless, despite the efforts made globally in predicting the most appropriate influenza strains to be included in the vaccine, occasionally, the influenza vaccine offers lower-than-expected efficacy due to antigenic differences against the strains circulating in the community [21].

As a consequence of this, it is important to sustain ongoing global influenza surveillance and vaccine effectiveness (VE), and promote influenza vaccination [22,23].

Sicily, the fifth most populous region in Italy, with around 5 million inhabitants, participates yearly in the national surveillance network RespiVirNet [24] by collecting respiratory specimens from patients presenting influenza-like illness (ILI) or severe acute respiratory infection (SARI). In addition to standardized clinical and epidemiological data, it provides key insights into disease transmission within the community.

In this context, a mid-term estimate of influenza VE was carried out in Sicily during the last season, 2023–2024 [25], documenting an unexpectedly high proportion of laboratory-confirmed influenza cases among vaccinated subjects (12.5%) caused by the predominant A(H1N1)pdm09.

Furthermore, a moderate VE (47.8%) was calculated, even lower than reported in other interim analyses from the Northern Hemisphere [23,26,27].

If, on one hand, influenza A(H3N2) viruses have typically been associated with reduced VE due to antigenic mismatch [28,29,30]; on the other hand, such evidence is less commonly documented for A(H1N1)pdm09.

Therefore, we have hypothesized that some drifted A(H1N1)pdm09 viruses may have circulated in Sicily, contributing, at least partly, to the low VE observed in the early stages of the season 2023–2024.

For this purpose, besides the real-time RT-PCR assays conventionally performed for influenza detection and subtyping in the seasonal surveillance activities, whole-genome sequencing (WGS) was retrospectively implemented to provide more detailed data on the evolution, diversity, and phylogenetic relationships of circulating influenza viruses. The WGS approach enables tracking changes across all influenza segments, providing a better understanding of evolutionary dynamics but also allowing for the exploration of potential associations between mutations across the viral genome and patient data in influenza surveillance [31,32].

In this study, we describe the epidemiology of influenza viruses during the season 2023–2024 in Sicily, supplemented by information obtained through WGS of a representative number of influenza-positive specimens randomly selected by a week of sampling.

## 2. Materials and Methods

### 2.1. Study Population and Case Definition

Respiratory specimens were submitted for testing as part of the routine influenza surveillance in Sicily within the national network RespiVirNet (formerly InfluNet; https://respivirnet.iss.it/default.aspx (accessed on 17 October 2024). From October 2023 to April 2024, patients of any age who met the case definition of ILI or SARI [33,34] were included.

A case of ILI was defined as a person presenting a sudden and rapid onset of at least one of the following systemic symptoms: fever or feverishness, malaise, headache, myalgia, and at least one of the respiratory symptoms: cough, sore throat, or shortness of breath. A case of SARI was defined as a patient with an acute respiratory infection that required hospitalization.

Samples from ILI patients were obtained with the collaboration of general practitioners and family paediatricians, who were preliminarily selected as “sentinel physicians”, operating within the RespiVirNet framework.

At the moment of sample collection, physicians were asked to complete a structured form including the following information: date of sampling, patient’s initials, date of birth, sex, date of onset of symptoms, underlying medical conditions, respiratory complications, and hospitalization (if required).

All specimens were conferred to the Regional Reference Laboratory of Sicily, operating at the University Hospital “Paolo Giaccone” of Palermo (Italy), and stored at −80 °C until further use.

### 2.2. Determination of Influenza Subtype, Amplicon-Based Whole-Genome Sequencing (WGS), and Assembly

According to the manufacturer’s instructions, viral nucleic acids were extracted from specimens using a QIAmp Viral RNA extraction kit (QIAGEN, Hilden, Germany). Extraction performance was verified by means of a one-step real-time (rt)-PCR assay targeting the housekeeping gene human ribonuclease P (RNase P) and then tested for influenza A and influenza B using a duplex one-step real-time retrotranscription (RT) assay. A test was considered positive when its cycle threshold (Ct) value was <40.

Moreover, influenza-positive samples were further analyzed for genetic subtyping using the protocols recommended by the national network RespiVirNet (all primer and probe sets are listed in Appendix A).

All rt-RT-PCR assays were performed with a QuantStudio^TM^ 7 Flex Real-Time PCR System (Applied Biosystems, Waltham, MA, USA).

To investigate the genetic profile of circulating influenza viruses which predominated in Sicily during the season 2023–2024, a representative subset of laboratory-confirmed cases subtyped as influenza A(H1N1)pdm09, selected according to the weekly sampling and with an rt-RT-PCR Ct value ≤30, was considered suitable for whole-genome sequencing.

To this purpose, viral RNA was first transcribed and amplified using a multi-segment reverse transcription-PCR (M-RTPCR) approach that simultaneously amplifies the eight segments of influenza A genome, irrespective of virus subtype. To this end, viral RNA underwent RT-PCR reactions targeting the universal termini of influenza A genome segments, using one set of three oligonucleotide primers, as proposed by Zhou et al. [35]. The adapted protocol is described in Appendix A. The amplicons obtained from M-RTPCR were then treated using ExoSAP-IT (Affymetrix, Santa Clara, CA, USA) at 37 °C for 15 min and analyzed with the Agilent Tapestation for the evaluation of both the PCR specificity and yield.

The purified amplicons were quantified using the Qubit dsDNA quantitation HS assay kit (Invitrogen, Waltham, MA, USA) and then transformed to DNA libraries using the Ion Xpress Plus Fragment Library Kit (Thermofisher, Waltham, MA, USA), following the manufacturer’s protocol.

Barcoded libraries were then loaded on E-gel Size Select™ II agarose gel (Invitrogen, USA) to collect fragments of 200–300 bp in length. Finally, libraries were further verified and quantified with the Agilent Tapestation and then sequenced on an Ion GeneStudio S5 Prime System (IonTorrent, Thermofisher).

Coverage analysis read mapping and assembly of the sequencing reads into target whole-genome nucleotide consensus sequences were performed with CLC Genomics Workbench version 23.0.5 (QIAGEN Digital Insights, Aarhus, Denmark). All influenza sequences were deposited on the Global Initiative on Sharing Avian Influenza Data (GISAID) database (accession numbers are reported in Appendix A).

### 2.3. Phylogenetic Analysis

For both HA and NA nucleotide sequences, preliminary analyses were carried out by using Nextclade tool version 3.7.4, available online (https://clades.nextstrain.org (accessed on 17 October 2024)), to assign the clade to each Sicilian influenza sequence and to explore the most probable phylogenetic placement. Therefore, influenza sequences collected in Italy during the season 2023–2024 were retrieved from the GISAID database and used in constructing phylogenetic trees, together with all WHO-recommended A(H1N1)pdm09-like vaccine strains.

To this purpose, each sequence dataset was aligned with MAFFT version 7, available online (https://mafft.cbrc.jp/alignment/server/index.html (accessed on 17 October 2024)), and the Neighbour-Joining method implemented in the MEGA X package was used for reconstructing phylogenetic trees from evolutionary distance data [36]. The tree topology’s reliability was estimated using the bootstrap re-sampling method with 1000 replicates. The best-fit evolutionary model and parameters were calculated, and the Tamura-Nei model of nucleotide substitution (TN93 + G + I) was estimated as the most appropriate for the datasets. Finally, the phylogenetic trees were visualized and annotated using FigTree v1.4.4.

### 2.4. Analysis of Deduced Amino Acid Sequences and Mutations

Deduced amino acid (AA) sequences were predicted with standard genetic codes and were aligned to the A/Victoria/4897/2022, recommended by the WHO as the influenza A(H1N1)pdm09 egg-based vaccine strain for the season 2023–2024 in the Northern Hemisphere (GISAID: EPI_ISL_17830834; GenBank: OQ718989, HA; OQ718988, NA). For the HA protein, AA residue numbering was carried out according to the scheme proposed by Burke et al. [37]. AA positions that differed from the prototype strain were defined as mutated, and AA variability was reported as the percentage of occurrence of each mutated AA at a given residue position.

### 2.5. Prediction of N-Glycosylation Sites and Vaccine Efficacy

The amino acids sequences were evaluated for potential N-glycosylation (Asn-X-Ser/Thr) sites using NetNGlyc 1.0 server [38].

Moreover, the antigenic relatedness of the influenza strains circulating in Sicily during the season 2023–2024 to the strain included in the vaccine was assessed by comparing the Sicilian HA protein sequences to that of the vaccine mentioned above the reference strain. Therefore, the epitope model proposed by Deem and colleague [39] was used to predict the vaccine efficacy, taking into account the epitopes A-E of the A(H1N1)pdm09 strains in analogy to the A(H3N2) virus. In more detail, the largest *p*-value estimated for each of the five antigenic sites, each calculated as the ratio between the number of mutated amino acids and the total number of amino acids in that specific epitope, defined both the epitope and the dominant epitope. For A(H1N1)pdm09, the association of vaccine efficacy and *p*_epitope_ is determined by the mathematical formula E = (−1.19 × *p*_epitope_ + 0.53) × 100, in which efficacy is 53% when the *p*_epitope_ = 0 (conserved epitope: a perfect match between the circulating strain and the vaccine strain). Vaccine efficacy was predicted considering the dominant epitope.

### 2.6. Statistical Analysis

The study population was arbitrarily subdivided into six different age groups: two for children, as defined by the Italian health system (≤14 years: ≤4 and 5–14 years old) and four for adults/elderly (15–24, 25–44, 45–64, and ≥65 years old).

Descriptive statistics were used to summarize the socio-demographic and clinical data of the study populations and viral characteristics. Frequency analyses for categorical variables were described with percentages. Comparisons of categorical variables were conducted using Pearson’s chi-square test, and *p*-values of 0.05 or less were considered statistically significant.

Time series data were considered for evaluating seasonal variations and periodic changes in the prevalence of influenza types/subtypes.

Data were processed with the STATA MP statistical software package v16.1 for Apple™ (StataCorp LLC, College Station, TX, USA).

### 2.7. Ethical Review

This laboratory-based study was carried out in full compliance with the rules concerning the protection of personal data adopted in Italy and satisfied the Helsinki Declaration. Since patients were tested for influenza as part of clinical management and for the national influenza surveillance programme, their consents were acquired at the time of their medical examinations. All data were analyzed anonymously. The study was approved by the institutional ethics committee of the University Hospital of Palermo (Italy) with approval number 09/2023.

## 3. Results

### 3.1. Seasonal Surveillance Data

During the surveillance season 2023–2024, 3175 respiratory specimens were collected from patients presenting ILI or SARI (Table 1). In total, 19.9% (n = 631) tested positive for influenza; among these, 89.5% (n = 565/631) were influenza A and 10.5% (n = 60/631) were influenza B.

Most of the influenza A-positive samples were subtyped as A(H1N1)pdm09 (95.8%; n = 541/565), whereas A(H3N2) was found in 4.2% (n = 24/565). While A(H1N1)pdm09 prevailed throughout the season, A(H3N2) was sporadically detected between weeks 50/2023 (11–17 December) and 10/2024 (4–10 March). Conversely, influenza B viruses began to circulate in the last part of the season, becoming the only influenza type detected since week 17/2024, all belonging to the Victoria lineage (Figure 1).

According to the Italian RespiVirNet operative protocol [24], the surveillance period officially runs from week 46/2023 (13–19 November) to week 17/2024 (22–28 April).

Notably, in Sicily, influenza viruses spread very early during the last season. They continued circulating in the community beyond the end of the reference period, as specified by the RespiVirNet operative protocol (Figure 1).

Given the structure of the surveillance organization in Sicily, almost all specimens were collected from community patients (Table 1). The study population mainly consisted of young individuals ≤14 years of age (73.1%; n = 2321/3175), among which the most significant part of reported influenza B cases were identified. In general, the prevalence of infection ranged between 13.1% and 26.7%, according to the age classes considered; no influenza B cases were identified from the age of 45 years.

There was a balanced distribution by sex (M:F ratio = 1.02). No difference was observed in the relative proportion of confirmed cases, except for influenza B, which significantly prevailed among males (14.8% vs. 6.4%).

In the overall study group, less than 20% of subjects were vaccinated for the season 2023–2024, although, as expected, the proportion was higher among the elderly, reaching 46.2%. The prevalence of infection was significantly higher among non-vaccinated subjects (21.7%); nevertheless, 11.3% (n = 64/564) of patients who received the vaccination acquired the infection.

Chronic conditions were documented in 10.8% (n = 342/3175). Among these patients, only influenza A infections were most often found in elderly adults.

Finally, respiratory complications were documented in 6.9% of the study population, and no significant differences emerged in the proportion of influenza types between patients with or without complications. Notably, the prevalence of complicated cases was the highest among patients aged 65 years and above (15.4%).

### 3.2. Influenza A(H1N1)pdm09 Genetic Characterization

To investigate the genetic changes and the phylogenetic relationships among the recently circulating influenza A(H1N1)pdm09 viruses in Sicily, the phylogenetic trees of both HA and NA genes were constructed, having a higher propensity to rapidly evolve because of the immune selective pressure.

As depicted in Figure 2, phylogenetic analysis of HA showed that all A(H1N1)pdm09 strains belonged to the genetic clade 6B.1A.5a.2a and its descendant 6B.1A.5a.2a.1 (hereafter defined as “5a.2a” and “5a.2a.1”, respectively), according to the Nextstrain classification.

In more detail, the 5a.2a cluster, which was represented by the 2023 vaccine strain for the Southern Hemisphere (A/Sydney/5/2021), has undergone further evolution to yield the three subclades C.1 (n = 6), C.1.8 (n = 11), and C.1.9 (n = 12). Moreover, within the clade 5a.2a.1, five different subclades were recognized: C.1.1, including the cell culture-based 2023–2024 vaccine strain for use in the Northern Hemisphere (A/Wisconsin/67/2022), and a single Sicilian strain, D.2 (n = 2), whereas two larger clusters further diverged to form subclades D.1 (n = 28) and D.3 (n = 12), together with the egg-based vaccine strain A/Victoria/4897/2022, recommended for the last season. Only one sequence from Sicily was classified as subclade D, strictly related to this latter vaccine strain (Figure 2).

Notably, no specific temporal patterns emerged within clades/subclades, thus suggesting a concurrent spread of viral variants in our geographic area.

Considering the phylogenetic tree built for the NA gene (Figure 3), two large diverging clusters were identified. The first one included the subclades C.5.3 (n = 6), C.5.3.1 (n = 21), and C.5.3.2 (n=2), whereas a second cluster was further subdivided into C.5.2 (n = 32) and C.5.1.1 (n = 7), to which belonged the three strains globally recommended in 2023–2024 vaccines.

Taken together, our findings suggest that A(H1N1)pdm09 strains circulating in Sicily during the season 2023–2024, similarly to what was observed in other Italian regions, have partially genetically drifted from the WHO’s recommended vaccine strains.

Figure 4 represents the lollipop plot of mutations recognized in the amino acid hemagglutinin-deduced sequences of A(H1N1)pdm09 strains in Sicily. The AA change R233Q was found in most strains (98.6%), while about half of the strains bore the substitutions R45K, T120A/E/I, A216T, E260D, and A277T, as well as D356E and H451N in the HA2 subunit. Additionally, some mutations were identified within the major antigenic sites. This is the case for K154R and G155R in Sa; K169Q in Ca1; and S137P, A139D/T, A141V, and R142K in Ca2, of which S137P and R142K were the most represented, being respectively found in 36.1% and 40.3% of A(H1N1)pdm09-like viruses. No variations were documented in the epitopes Sb and Cb.

The major antigenic sites are defined according to Martinez JL et al. [40] and depicted with coloured rectangles. AA-mutated residues are numbered according to Burke DF et al. [37]. The proportion of A(H1N1)pdm09 Sicilian strains harboring the reported AA mutation is shown next to the label as a percentage. Except AA mutations falling within the major antigenic sites (indicated in red circles), only mutations with frequencies ≥10% are reported.

The impact of these mutations on vaccine efficacy was evaluated using the *p*_epitope_ model proposed by Deem and colleague [39] against the A(H1N1)pdm09 strain included in the 2023–2024 vaccine for the Northern Hemisphere (Table 2).

Considering the five main antigenic sites of the HA protein, the maximum *p*_epitope_ value (0.125) was calculated in epitope A (thus defined as the dominant epitope), based on a subset of 23 HA sequences, all referring to the subclades C.1.8 and C.1.9 (Figure 2). The computed value suggested that the worst-case vaccine efficacy against these viruses would be 71.92% (E = 38.12% of 53%, *p*_epitope_ = 0), because of a genetic drift from the vaccine strain A/Victoria/4987/2022. With regard to the other four epitopes (from B to E), the weighted mean VEs (53%) ranged between 48.73% and 52.85% (Table 2).

Finally, the prediction analysis of N-linked glycosylation sites evidenced that all but two strains from Sicily carried seven potential motifs in HA (HA1 position: 10, 23, 87, 162, 276, 287; and HA2: 154). Only two HA sequences, both belonging to the subclade C.1.9 (A/Palermo/106432/2023 and A/Siracusa/108015/2024), harboured the mutation T278N potentially responsible for the loss of the site of glycosylation. However, this modification did not fall within the main epitopes.

In this study, the mutation analysis of NA sequences showed limited evolution as compared with the vaccine strain A/Victoria/4897/2022. Only for seven AA mutations the frequency of detection exceeded 30%. At the same time, all other AA substitutions were sporadically found across the NA sequence, very likely due to the natural evolution of influenza viruses (Figure 5). Two pairs of amino acids, I241L + S339L and D50N + E382G, were found in 47.1% and 89.7% of strains, respectively.

The major antigenic sites are defined according to Martinez JL et al. [40] and depicted with colored rectangles. AA mutated residues are numbered according to Burke DF et al. [37]. The proportion of A(H1N1)pdm09 Sicilian strains harboring the reported AA mutation is shown next to the label as a percentage. Only mutations with frequencies ≥10% are reported.

Sicilian strains generally carried eight potential N-glycosylation sites in NA positions 42, 50, 58, 63, 68, 88, 146, and 235. In two strains (A/Enna/105929/2023 and A/Enna/106169/2023: subclade C.5.3.2), the N73S substitution contributed to an increase in the number of sites (position 71), whereas a small group of sequences, all belonging to the subclade C.5.1.1 and sharing the amino acid D50 with the vaccine strain, lost one potential N-glycosylation site at position 50.

None of the analyzed NA sequences presented the H275Y mutation associated with oseltamivir resistance, nor the I223R reported in zanamivir-resistant variants. Only one strain bore the substitution S247N reported to be responsible for reduced susceptibility to neuraminidase inhibitors.

Moreover, all the functional residues of the active site in the head domain of NA (R118, D151, R152, R224, E276, R292, R371, and Y406), as well as the ten structural “framework residues” (E119, R156, W178, S179, D198, I222, E227, N294, and E425) were conserved among all Sicilian strains.

Finally, to further investigate the genomic changes in the viruses collected in Sicily during the season 2023–2024, AA sequences of the remaining six segments (PB2, PB1, PA, NP, MP, and NS) of the influenza genome were compared to the corresponding sequence of the reference strain (Appendix A). The analysis revealed several polymorphisms at very low frequency. In contrast, some AA substitutions were found in most of the sequenced strains (i.e., Q453P in the nucleoprotein, S212P in the non-structural protein 1, and G67E/K in the non-structural protein 2). No AA changes were detected at high frequency in the matrix protein (segment 7).

## 4. Discussion

Influenza surveillance is essential for investigating the epidemiology of the virus and determining the composition of the seasonal vaccine.

Sicily participates in the national respiratory infection surveillance network, seasonally collecting prospective epidemiological and molecular data.

This study aimed to describe the key points of the surveillance season 2023–2024 in Sicily (Italy) and to explore the molecular characteristics of the predominantly circulating A(H1N1)pdm09 influenza virus. The genomic analysis of influenza viruses circulating from October 2023 to April 2024 is reported, adding consistency to the limited number of 2023–2024 complete genetic sequences available on GISAID from our country, which almost all originating from Northern Italy. Whole-genome sequencing was applied to investigate their phylogenetic relationships and to evaluate how deduced amino acid changes may have affected the effectiveness of seasonal vaccines.

After disruption in the viral circulation observed in 2020–2021 with the emergence of SARS-CoV-2, the spread of influenza has returned to the pre-pandemic levels, as globally documented in the last two years [41]. In Sicily, the outbreak of 2023–2024 was characterized by the early spread of influenza, even before the surveillance time period planned at the national level [24]. In accordance with data communicated by the ECDC [42], A(H1N1)pdm09 predominated over the other type/subtypes.

In our study population, most of the subjects presenting ILI or SARI were unvaccinated. Nevertheless, it is worth noting that about 12% of individuals who had received the 2023–2024 vaccine against influenza were laboratory-confirmed cases.

Although this is an expected event, especially in seasons in which A(H3N2) is the prevalent subtype, the rate of infection among vaccinated individuals is usually much lower when A(H1N1)pdm09 viruses predominate [25,43,44].

In this context, mid-term analyses annually performed in the early stages of the surveillance season provide further data by estimating the vaccine effectiveness in the community [45].

VE estimates against A(H1N1)pdm09 reported over the previous years ranged from 33% to 75% [46,47,48,49], thus placing our current season’s overall estimate of 43.8% [25] at roughly mid-range. However, the VE reported from Sicily was modest and lower than that reported in 2023–2024 from Canada [23,30] or other European countries [27,50].

For this reason, the circulation of A(H1N1)pdm09 viruses potentially drifted from the vaccine strains has been hypothesized in our geographic area.

At the global level, influenza A viruses continuously evolve across different seasons, resulting from drift in their phylogeny and accumulating genomic modifications that may affect the efficacy of vaccine-based preventive measures.

Consequently, the genetic characterization of circulating viruses is of paramount importance, and in a modern vision of surveillance in public health, whole-genome sequencing surely represents an invaluable tool.

For this purpose, WGS was carried out in Sicily on a representative subset of A(H1N1)pdm09 viruses identified throughout the surveillance period.

Phylogenetic analyses of the HA and NA genes substantially confirmed the WHO/ECDC surveillance report results for the European region [42]. For the HA sequences, the phylogenetic tree revealed two main separated clusters, the first of which included a larger subset of sequences (about 60%), including the vaccine strain A/Victoria/4897/2022, all attributable to the clade 5a.2a.1 predominant in the Americas, Japan, and some countries in Europe [42]. Alongside this, a second group of sequences clustered in the clade 5a.2a, together with the strains reported from other regions of Italy. Notably, this cluster was represented in a minor proportion in Europe [42] and included those sequences harboring the AA mutations in epitope A of the hemagglutinin responsible for the lowest predicted VE (53%) reported in this study.

The antigenic variations on the influenza virus reflect the accumulation of substitutions gradually acquired over time, specifically on the HA protein, resulting in the genetic distance between the predominant circulating strains and that been included in the seasonal formulation. Therefore, it leads to reduced VE and the need to re-evaluate the composition of the vaccine, which undoubtedly still represents the best preventive tool against influenza.

In this study, the identification of a large cluster of Sicilian sequences not belonging to the clade where the vaccine strain A/Victoria/4897/2022 may explain, at least in part, the low vaccine efficacy predicted through the *p*_epitope_ model.

Several AA changes were found in the HA sequences, some falling within the major epitopes. The most represented AA substitution was Q223R, which has become dominant since 2009 [51]. It is reported to be an egg-adapted change in the HA head domain of A(H1N1)pdm09 viruses, able to promote virus replication in eggs, alter antigenicity, and influence immune response [52,53].

Instead, slightly more than 40% of sequences, all clustering within the subclade 5a.2a, bore the substitution T16A matched to R142K. The latter, as opposed to the AA substitution named R142K (Ca2) reversion, emerged in egg-derived strains representative of both inactivated and live attenuated influenza vaccines for the season 2023–2024 [30].

It is well known that neuraminidase plays a relevant role in viral infectivity, and the stability of the enzyme’s active site is important for NA’s activity. It is widely reported that this protein undergoes less pronounced antigenic variability [54], considering both the framework residues and the functional residues of the catalytic site involved in the cleavage of sialic acids, enabling the release of new virion progeny [55].

Nevertheless, some specific AA mutations may occur and interfere with the efficacy of the most common antiviral drugs active against influenza A viruses, such as oseltamivir and zanamivir [56,57,58]. Compared to the vaccine strain for the season 2023–2024, the influenza strains analyzed in this study did not show any AA changes in the catalytic site or related to NA inhibitor resistance, confirming the observation of other authors [59].

It is worth noting the lack of both H275Y and I233R in our study population. This is very likely due to the almost non-existent use of antiviral agents and, thus, the absence of positive selective pressure in our community [60]. These results indicated that NA inhibitors would still have a good therapeutic effect against influenza viruses circulating in Italy.

This study has a set of limitations and strengths. Younger individuals ≤14 years of age were prevalent in our population setting. This inevitably introduced a selection bias, as well as the sampling source bias, based mainly on the community. This latter aspect, on one hand, may have introduced further potential biases; on the other hand, it constitutes a good picture of circulating viruses in the general population.

However, it should be stressed that our findings referred to data collected from a single region in southern Italy, even though the seasonal surveillance involved sentinel physicians operating in all nine provinces of Sicily, ensuring adequate coverage of the whole area.

Moreover, despite the number of A(H1N1)pdm09 cases subjected to WGS covering the whole surveillance season and providing relevant information on the molecular epidemiology of influenza in Sicily, it cannot exclude different patterns among uncharacterized strains.

Finally, it should be highlighted that the estimates for potential vaccine efficacy were simply assessed based on the mathematical model adopted. It is worth recalling that the *p*_epitope_ model was developed to support researchers and health authorities in the quantification of the antigenic distance between dominant circulating strains and candidate vaccine strains, supplementing the more time-consuming and more expensive hemagglutinin inhibition assay, which remains the gold standard in the evaluation of vaccine efficacy.

In conclusion, this study reported the circulation of drifted influenza viruses harboring AA mutations, some of which are potentially related to low vaccine effectiveness. This evidences the need for real-time genomic surveillance of contemporary influenza viruses and its implications for public health decisions.

The application of whole-genome sequencing in the seasonal surveillance provides key insights into the evolution, diversity, and phylogenetic relationships of influenza viruses, but also represents an invaluable tool in exploring the epidemiology of the disease and transmission within communities, as well as in association studies linking the viral genetic background to patients’ clinical data.

## Figures and Tables

**Figure 1 viruses-16-01644-f001:**
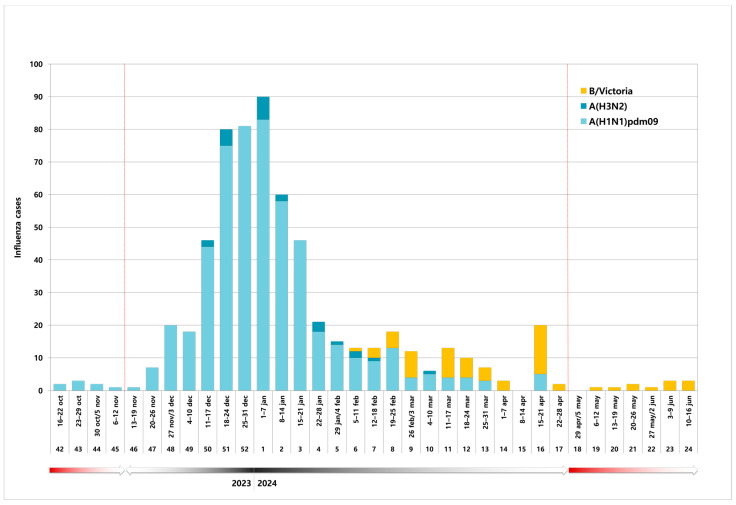
Weekly distribution of influenza A and B laboratory-confirmed cases from October 2023 to June 2024 in Sicily (Italy).

**Figure 2 viruses-16-01644-f002:**
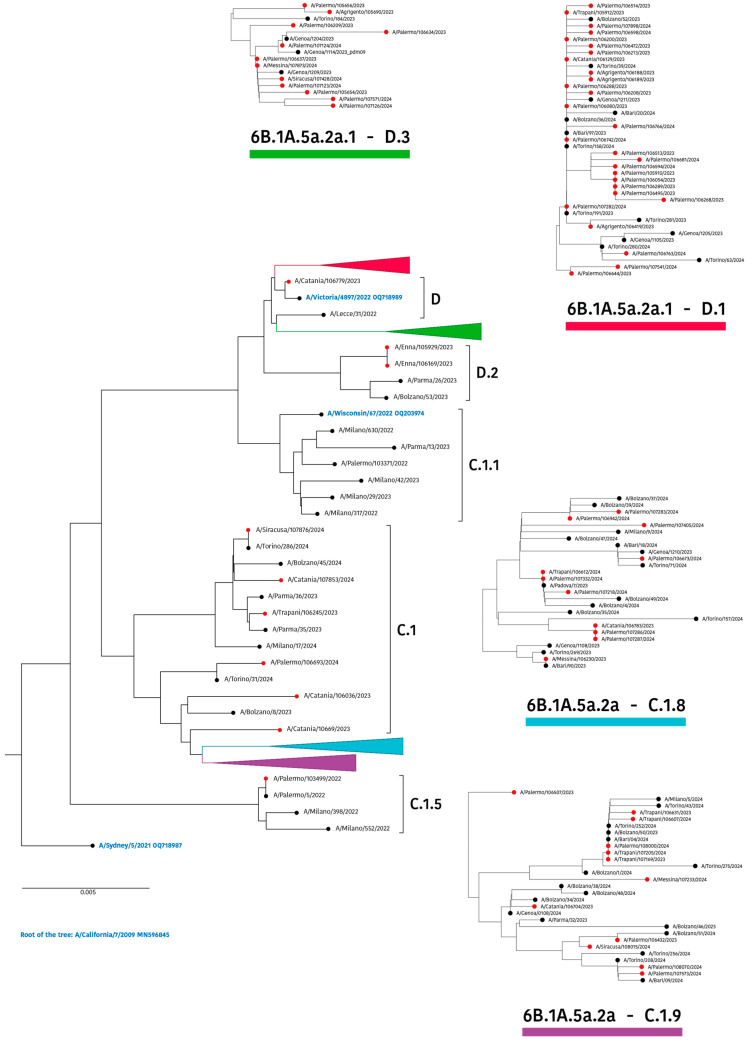
Neighbour-joining phylogenetic tree of HA nucleotide sequences of influenza A(H1N1)pdm09 strains collected in Sicily between October 2023 and April 2024. Solid red circles indicate the study sequences, while A(H1N1)pdm09-like vaccine strains are shown in blue. The strain A/California/7/2009 (GenBank: MN596845) was used as an outgroup. Clades and subclades are defined according to the Nextstrain classification.

**Figure 3 viruses-16-01644-f003:**
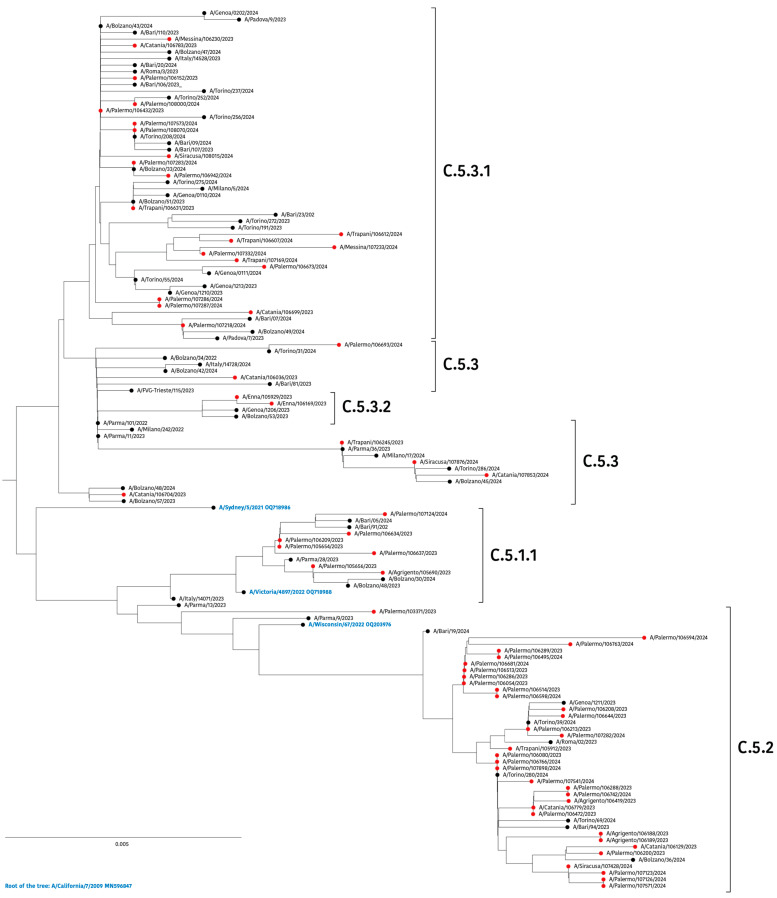
Neighbour-joining phylogenetic tree of NA nucleotide sequences of influenza A(H1N1)pdm09 strains collected in Sicily between October 2023 and April 2024. Solid red circles indicate the study sequences, while A(H1N1)pdm09-like vaccine strains are shown in blue. The strain A/California/7/2009 (GenBank: MN596847) was used as an outgroup. Clades and subclades are defined according to the Nextstrain classification.

**Figure 4 viruses-16-01644-f004:**
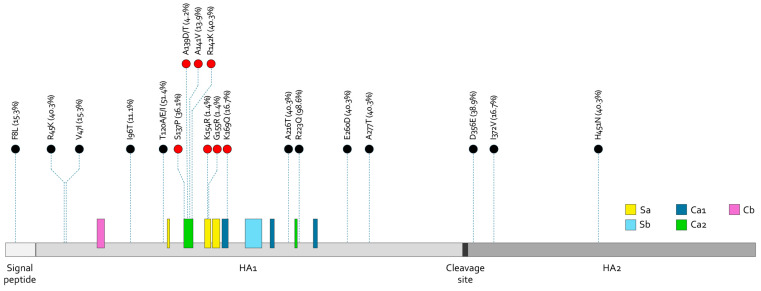
Lollipop plot of amino acid substitutions in the HA protein of Sicilian A(H1N1)pdm09 circulating during the season 2023/2024, in comparison to the seasonal vaccine strain for the Northern Hemisphere (A/Victoria/4897/2022).

**Figure 5 viruses-16-01644-f005:**
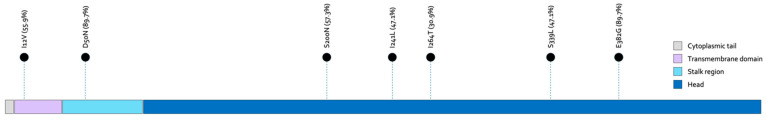
Lollipop plot of amino acid substitutions in the NA protein of Sicilian A(H1N1)pdm09 circulating during the season 2023/2024, in comparison to the seasonal vaccine strain for the Northern Hemisphere (A/Victoria/4897/2022).

**Table 1 viruses-16-01644-t001:** Demographic and clinical characteristics of patients presenting ILI/SARI confirmed influenza cases, and relative proportions attributable to influenza A and B virus in Sicily. Surveillance season: 2023–2024 (% by row).

Demographic Characteristic	Total Subjects	Influenza Cases	Influenza A	Influenza B
Study population [n (%)]	3175	631 (19.9)	565 (89.5) *	66 (10.5)
Healthcare setting [n (%)]				
Community-based (ILI)	3109 (97.9)	603 (19.4)	545 (90.4)	58 (9.6)
Hospital-based (SARI)	66 (2.1)	28 (42.4)	20 (71.4)	8 (28.6)
Age-group [years; n (%)]				
Children (≤14 years)	2321 (73.1)	473 (20.4)	415 (87.7)	58 (12.3) **
Adults (>14 years)	854 (26.9)	158 (18.6)	150 (94.9)	8 (5.1)
≤4	1379 (43.4)	221 (16.0)	203 (91.8)	18 (8.1)
5–14	942 (29.7)	252 (26.7)	212 (84.1)	40 (15.9)
15–24	82 (2.6)	18 (21.9)	15 (83.3)	3 (16.7)
25–44	179 (5.6)	40 (22.3)	35 (87.5)	5 (12.5)
45–64	281 (8.9)	59 (21.0)	59 (100.0)	0
≥65	312 (9.8)	41 (13.1)	41 (100.0)	0
Sex [n (%)]				
Female	1569 (49.4)	327 (20.8)	306 (93.6)	21 (6.4)
Male	1606 (50.6)	304 (18.9)	259 (85.2)	45 (14.8) ***
Vaccination [n (%)]				
No	2611 (82.2)	567 (21.7)	504 (88.9)	63 (11.1)
Yes	564 (17.8) ^#^	64 (11.3)	61 (95.3)	3 (4.7)
Comorbidities [n (%)]				
No	2833 (89.2)	578 (20.4)	512 (88.6)	66 (11.4)
Yes	342 (10.8)	53 (15.7)	53 (100.0)	0
One co-morbidity	197 (57.6)	25 (12.8)	25 (100.0)	0
Multiple co-morbidities	145 (42.4)	28 (19.9)	28 (100.0)	0
Respiratory complications [n (%)]				
No	2956 (93.1)	607 (20.5)	544 (89.6)	63 (10.4)
Yes	219 (6.9)	24 (11.0) ^##^	21 (87.5)	3 (14.3)

* A(H1N1)pdm09: 95.8% (n = 541/565); A(H3N2): 4.2% (n = 24/565). ** *p*-value = 0.01, ≤14 years vs. >14 years. *** *p*-value = 0.01, males vs. females. # ≥65 years old: 46.2%. ## ≥65 years old vs. <65 years old: 15.4% vs. 3.0%.

**Table 2 viruses-16-01644-t002:** Number of mutations found on the main epitopes of the hemagglutinin protein among Sicilian A(H1N1)pdm09 strains circulating during the season 2023–2024 and computed prediction of vaccine efficacy.

Vaccine Strain	Epitope	Number of Mutations	Number of Strains	*p* _epitope_	Vaccine Efficacy (VE) *	
					53%	100%
A/Victoria/4987/2022	A	0	28	0.000	53.00	100.00
		1	6	0.042	48.00	90.57
		2	15	0.083	43.12	81.36
		3	23	0.125	38.12	71.92
				*Weighted mean* **	*45.76*	*86.34*
	B	0	70	0.000	53.00	100.00
		1	2	0.046	47.53	89.68
				*Weighted mean* **	*52.85*	*99.71*
	C	0	14	0.000	53.00	100.00
		1	43	0.030	49.43	93.26
		2	14	0.061	45.74	86.30
		3	1	0.091	42.17	79.57
				*Weighted mean* **	*49.28*	*92.98*
	D	0	1	0.000	53.00	100.00
		1	40	0.021	50.50	95.28
		2	11	0.042	48.00	90.57
		3	19	0.063	45.50	85.85
		4	1	0.083	43.12	81.36
				*Weighted mean* **	*48.73*	*91.94*
	E	0	39	0.000	53.00	100.00
		1	18	0.029	49.55	93.49
		2	12	0.059	46.00	86.79
		3	3	0.088	42.53	80.25
				*Weighted mean* **	*50.55*	*95.37*

* VE (53%) was calculated from E = (−1.19 × *p*_epitope_ + 0.53) × 100, in which efficacy is 53% when the *p*_epitope_ = 0. VE (100%) = E (53%)/0.53. ** For each epitope, VE mean value has been weighted considering the number of strains included within each stratum.

## Data Availability

The data underlying this article are available in the article. If needed, please contact the corresponding author at the following email address: fabio.tramuto@unipa.it.

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
