# Peer review of "Insights into Genetic and Antigenic Characteristics of Influenza A(H1N1)pdm09 Viruses Circulating in Sicily During the Surveillance Season 2023–2024: The Potential Effect on the Seasonal Vaccine Effectiveness"

_viruses, 2024, doi:10.3390/v16101644_

Round 1
Reviewer 1 Report
Comments and Suggestions for Authors
The authors have presented of very interesting and actual manuscript. Unfortunately the present data not distinguished the viruses which were received from vaccinated persons. Therefore the manuscript need some improvement:
1. It should be collect in additional table all viruses from vaccinated persons with ID number in GISAID database.
2. Indicate every virus as belonge to reference strain (A/Sydney/5/2021 or A/Victoria/4897/2022 or other) according the genetic characteristics.
3. Analyse the functional mutations in each virus which may influence to virus characteristics. Perhabs it allow to find common mutations in the viruses which were received from vaccinated persons. 4. In separate graph have to indicate the age of patient.
5.It should to analysed the age of vaccinated patients with influenza (perhaps they have 5 years - the age with particularity of immune system).
Author Response
TO REVIEWERS
We sincerely thank the Reviewers for providing us this opportunity to further revise our manuscript.
A point-by-point description of how each comment was addressed in the manuscript is given below. Original reviewers' comments in boldface, responses in regular typeface.
To Reviewer #1
The authors have presented of very interesting and actual manuscript. Unfortunately the present data not distinguished the viruses which were received from vaccinated persons. Therefore the manuscript need some improvement:
- It should be collect in additional table all viruses from vaccinated persons with ID number in GISAID database.
Response
According to the Reviewer’s suggestion, two columns have been added in Table S3 including age and vaccination status of patients.
- Indicate every virus as belonge to reference strain (A/Sydney/5/2021 or A/Victoria/4897/2022 or other) according the genetic characteristics.
Response
Thank you for asking for further elucidation on this specific aspect. Figure 2 and 3 respectively depict the NJ phylogenetic trees of HA and NA nucleotide sequences of influenza A(H1N1)pdm09 strains collected in Sicily during the last surveillance season.
The vaccine strains included in the 2023-2024 formulation for both the northern and southern hemispheres (A/Victoria/4897/2022, A/Sydney/5/2021, and A/Wisconsin/67/2022) have been reported. It is easily evident that none of Sicilian strains clustered with the strain A/Sydney/5/2021. Therefore, in our opinion, we do not consider useful to add further data on this topic.
- Analyse the functional mutations in each virus which may influence to virus characteristics. Perhabs it allow to find common mutations in the viruses which were received from vaccinated persons. 4. In separate graph have to indicate the age of patient.
Response
As better described in the answer to the following question, NGS sequencing was carried out on a random selection of A(H1N1)pdm09 positive samples, according to week of sampling. Consequently, the number of vaccinated patients was not proportionally identified. Due to the very few number of viral sequences obtained from this population setting, specific analyses could therefore not be performed.
- It should to analysed the age of vaccinated patients with influenza (perhaps they have 5 years - the age with particularity of immune system).
Response
We thank the Reviewer for the comment. However, we wish to emphasize that only 7 out of 77 sequenced influenza strains belonged to influenza vaccinated patients, of which only one was under 5 years old. Therefore, a specific analysis is not applicable in such a case. Nevertheless, as previously stated, both patient’s age and vaccination status have been included in Table S3.
Reviewer 2 Report
Comments and Suggestions for Authors
Influenza A and B viruses continue to represent a major respiratory threat worldwide. Due to constant evolution resulting from both antigenic drift and antigenic shift, vaccination efforts to control these viruses must be updated on an annual basis according to the predicted circulating strains. This is considered a non-exact science, as the efficacy of yearly influenza vaccines varies considerably. Although the spread of influenza was disrupted by the Covid-19 pandemic, it has now returned to pre-pandemic levels, undoubtedly aided, in part, by the generalized vaccine hesitancy that has taken hold. Indeed, in this study, less than 20% of study participants were vaccinated against influenza, although this percentage was significantly higher among the elderly.
In order to understand the evolution of influenza during this period of time, this study offers a retrospective examination of the variability of the predominant strain, A(H1N1)pdm09, circulating in Sicily during the most recent 2023-2024 influenza season. Importantly, the data consist of whole-genome sequences of the virus circulating in the region over this period of time. The major findings resulting from the study include: 1) the strains of the virus circulating have drifted significantly from the vaccine strain; 2) accumulated mutations in the HA protein were the most impactful differences between the vaccine and circulating strains; 3) some of the HA mutations, especially Q223R, were identified in major antigenic sites; 4) these differences resulted in diminished vaccine effectiveness (VE), compared to other European countries; 5) this resulted in the need to re-evaluate the components of the influenza vaccine moving forward; and, 6) although amino acid substitutions were found in other segments, variation at critical positions does not extend to the NA protein due to the lack of the use of antiviral agents directed at this protein.
This study is well-designed and appropriately interpreted. It makes a profoundly strong case for the use of whole-genome sequencing of seasonal influenza to monitor the evolution of the virus to more accurately and effectively inform vaccine composition. The study is considered to make a significant contribution to the field. There are no discerned weaknesses, beyond some improvements required in the use of the English language and grammar.
Comments on the Quality of English Language
Moderate editing of the use of the English language and grammar is required.
Author Response
To Reviewer #2
This study is well-designed and appropriately interpreted. It makes a profoundly strong case for the use of whole-genome sequencing of seasonal influenza to monitor the evolution of the virus to more accurately and effectively inform vaccine composition. The study is considered to make a significant contribution to the field. There are no discerned weaknesses, beyond some improvements required in the use of the English language and grammar.
Response
We are truly grateful for the appreciation of our paper. According to the Reviewer’s suggestion, the entire manuscript has been revised as best we can.
Reviewer 3 Report
Comments and Suggestions for Authors
Dear Authors,
You have carried out great research and have written an excellent article!
Only a few remarks.
1). In Figure 3, check, please, designation of clades C.5.3 (twice) and C.5.3.2, and their coinciding with text in lines 296-297.
2). Figure 4. It is difficult to read important information because of too small letters in Figure 4.
If possible, could you increase the font size, please.
3) Check, please, a sentence repeated twice in lines 322-324 and 362-364.
4) I have no doubt that you have written everything correctly in subsection 2.5 and Table 2, but I have not understood something, could you explain, please.
What does Pepitope mean? Are p-value (L. 193) and Pepitope (L.196) the same parameter or different? It is unclear from the text. What does designate E (L.196). Does it mean Vaccine efficacy (VE) or not?
There are two columns on the right in the Table 2, that are combined by common subtitle Vaccine efficacy (VE). What does mean each of the columns?
Author Response
TO REVIEWERS
We sincerely thank the Reviewers for providing us this opportunity to further revise our manuscript.
A point-by-point description of how each comment was addressed in the manuscript is given below. Original reviewers' comments in boldface, responses in regular typeface.
To Reviewer #3
Dear Authors,
You have carried out great research and have written an excellent article!
Only a few remarks.
As for the Reviewer 2, we are sincerely glad and grateful for the appreciation of our work. A point-by-point revision has been carried out according to the Reviewer’s suggestions, as shown in the following:
- In Figure 3, check, please, designation of clades C.5.3 (twice) and C.5.3.2, and their coinciding with text in lines 296-297.
Response
We would like to thank the Reviewer for his careful observation. While we understand the Reviewer's concerns and recognize the potential miscommunication in Figure 3, as currently depicted, the subclade C.5.3 has been reported twice according to the Nextstrain classification.
Conversely, the text in lines 296-297 has been revised because of some typos encountered.
- Figure 4. It is difficult to read important information because of too small letters in Figure 4.
If possible, could you increase the font size, please.
Response
Thank you for the suggestion. The font sizes have been increased in order to improve the readability.
- Check, please, a sentence repeated twice in lines 322-324 and 362-364.
Response
The sentences in lines 322-324 and 362-364 are part of the figure caption, which unfortunately have been incorporated in the text by mistake. Font size has been reduced according to the figure caption and the text clearly separated from the rest of the manuscript in order to avoid any misunderstanding during the layout phase. Again, we thank the Reviewer for his careful observation.
- I have no doubt that you have written everything correctly in subsection 2.5 and Table 2, but I have not understood something, could you explain, please.
What does Pepitope mean? Are p-value (L. 193) and Pepitope (L.196) the same parameter or different? It is unclear from the text. What does designate E (L.196). Does it mean Vaccine efficacy (VE) or not?
There are two columns on the right in the Table 2, that are combined by common subtitle Vaccine efficacy (VE). What does mean each of the columns?
Response
We sincerely apologize for the difficulties the Reviewer encountered in reading the text.
As described by Deem and colleague, a p-value was estimated for each of the five antigenic sites of the hemagglutinin, from A to E. Among these, the largest p-value identified the “dominant epitope” and, consequently, this p-value assumed the label “pepitope”.
Moreover, the term “E” reported in the equation E = (-1.19 x pepitope + 0.53) x 100 generically identifies the “vaccine efficacy”, whereas VE (predicted vaccine efficacy) is used when pepitope, included in the calculation, refers to the largest p-value found.
Finally, the two columns on the right in the Table 2 report the estimated vaccine efficacy calculated when:
- the pepitope=0 (a perfect match between the circulating strain and the vaccine strain, no mutations in the antigenic site / conserved epitope), assuming the term 0.53 estimated for A(H1N1)pdm09 (the highest efficacy for this virus) and, therefore labeled as VE(53%).
- VE(100%) is calculated by normalizing the VE(53%) to a theoretical 100% efficacy, again when pepitope=0.
We understand the Reviewer's concern and we agree that some concepts may be unclear. Nevertheless, we have chosen to strictly report the definition provided by Deem and colleague with the aim of preventing any different interpretation, remaining in agreement to other authors who have already addressed this topic (i.e. DOI: 10.1038/s41598-020-80895-w and DOI: 10.1002/jmv.25328).
We sincerely hope to have clarified this aspect and are available for further clarification. Some minor revisions have been applied to the manuscript.
For further details, please, see also the paragraph “Calculation of pepitope” at page 544 of Deem and colleague [Ref. 39].